# Bounds on Absolute Gypsy Moth (*Lymantria dispar dispar*) (Lepidoptera: Erebidae) Population Density as Derived from Counts in Single Milk Carton Traps

**DOI:** 10.3390/insects11100673

**Published:** 2020-10-03

**Authors:** Ksenia S. Onufrieva, Alexey V. Onufriev, Andrea D. Hickman, James R. Miller

**Affiliations:** 1Department of Entomology, Virginia Tech, Blacksburg, VA 24061, USA; anhickma@vt.edu; 2Department of Computer Science, Virginia Tech, Blacksburg, VA 24061, USA; alexey@cs.vt.edu; 3Department of Physics, Virginia Tech, Blacksburg, VA 24061, USA; 4Center for High End Computer Systems, Virginia Tech, Blacksburg, VA 24061, USA; 5Department of Entomology, Michigan State University, East Lansing, MI 48824, USA; Miller20@msu.edu

**Keywords:** gypsy moth, pheromone-baited trap, milk carton trap, trap efficiency, pheromone plume, trapping radius

## Abstract

**Simple Summary:**

Gypsy moth is one of the most devastating forest pests in the Eastern USA. In this paper, we derive a simple formula to interpret catches in monitoring moth traps deployed by management programs.

**Abstract:**

Estimates of absolute pest population density are critical to pest management programs but have been difficult to obtain from capture numbers in pheromone-baited monitoring traps. In this paper, we establish a novel predictive relationship for a probability (*spT_fer_(r)*) of catching a male located at a distance *r* from the trap with a plume reach *D*.
$spT_{fer}\left(r\right)=\left\{\begin{array}{r} \frac{spT_{fer}\left(0\right)}{1+{\left(\frac{r}{D}\right)}^{2}},\;r\leq R_{max}\\ 0,\;r>R_{max} \end{array}\right.$, where *spTfer*(0) is the probability of catching an insect located next to the trap and Rmax is the maximum dispersal distance for the insect during the trapping period. The maximum dispersal distance for gypsy moth is known to be 1600 m. The probability of catching a gypsy moth male located next to a United States Department of Agriculture (USDA) milk carton pheromone-baited trap is 0.37, the overall probability of catching a male from the entire trapping area (*T_fer_*) of ~800 ha is 0.0008, and plume reach of this trap is D = 26 ± 3 m. The equation for *spT_fer_(r)* is used to derive statistical upper and lower bounds (95% confidence interval) on the population density for the given value of a single trap catch. This combination of trap parameters appears to produce an effective trap: even a catch of 1 male provides meaningful lower and upper bounds on absolute population density. Applications in the management programs are discussed, and a look-up table is provided to translate the catches in USDA milk carton pheromone-baited traps to absolute population bounds, which can help design better management strategies.

## 1. Introduction

European gypsy moth (*Lymantria dispar dispar* (L.), Lepidoptera: Erebidae) is one of the most devastating forest pests in the Eastern United States. It was introduced to Medford, MA, USA in 1869 and since then has been continuously expanding its range. Gypsy moth larvae are extreme folivores feeding on leaves of over 300 different trees and shrubs [1,2]. Gypsy moth is primarily considered a forest pest; however, during outbreaks it can pose a threat to various fruit and nut crops, lead to reductions in residential property values as a result of defoliation, damage public greenspaces, and cause allergic reaction in humans [3,4,5]. Gypsy moth management efforts include outbreak suppression, slowing its spread in the transition zone, and eradication of populations that arrive outside of the invaded range. All of these management programs rely on traps baited with synthetic gypsy moth sex pheromone (+)-disparlure to detect gypsy moth populations, estimate population density, and evaluate success of applied treatments [6]. Trapping counts are favored over other measures of gypsy moth density such as egg mass and pupal counts because they are by far the least costly to obtain and were shown to be well correlated with egg mass and pupal counts [7,8,9].

Population density assessment is a critical part of any pest management program. Traps are utilized to detect and delimit small isolated insect populations and to estimate abundance and periods of activity [10,11,12,13,14,15,16,17,18]. Extensive research has been conducted over the years to establish the range of trap attraction, evaluate trap efficiency, estimate effective sampling area and catch probability, with the ultimate goal of interpreting trap catches and relating them to the actual population density [19,20,21,22,23,24,25]. Recent research by Bau and Cardé [26] demonstrated a high probability of false-negative trap catches when the density of an insect population is low and concluded that trap efficiency had a profound effect on detectability. Translating catch numbers from monitoring traps into estimates of absolute density has proven challenging for any insect [11,15,27]. The availability of statistically reliable estimates of the absolute density is critical for assessing efficacy of existing pest management programs and making improvements.

Recent explorations using computer simulations have provided substantial insight into the mechanics and meaning of a catch number in a pheromone-baited monitoring trap targeting insects foraging by random walks [28]. This approach treats insects much like diffusing molecules and a trap like a heat sensor recording particle hits from multiple distances [29]. This approach has been successful in translating catch numbers into estimates of absolute density for codling moth (*Cydia pomonella*, Lepidoptera: Tortricidae) in apple orchards [30], spotted wing drosophila (*Drosophila suzukii*, Diptera: Drosophilidae) [31], and brown marmorated stink bug (*Halyomorpha halys*, Hemiptera: Pentatomidae) [32].

In the research detailed below, we began the study by applying the existing framework developed by [28], which defines key parameters of the trap-insect system, such as plume reach and probability of catch. This analysis sets the stage for deriving and experimentally validating a simple mathematical relationship between the catch probability and distance to the trap, *spT_fer_(r)*. The novel relationship turns out to be instrumental in deriving mathematically rigorous statistical bounds of absolute population density for gypsy moth, which was the ultimate goal of this work.

## 2. Materials and Methods

The study was conducted in Appomattox-Buckingham State Forest, VA, USA, in summers of 2015, 2016, 2019, and 2020. The forest is planar and experiences shifting rather than a dominant wind direction. Laboratory-reared gypsy moth males were obtained as pupae from the USDA Animal and Plant Health Inspection Service, Pest Survey Detection and Exclusion Laboratory, OTIS Air National Guard Base, Buzzards Bay, MA, USA. Pupae were kept in laminated paper cups covered with mesh screening. Solvent red 26 dye (Royce International, Paterson, NJ, USA) was added to the larval diet at the rearing facility; it transferred into adults so as to allow clean differentiation between released and wild male moths. We released males by hand counting out the exact number of males at each release point.

Male moths were captured in standard USDA milk carton pheromone traps baited with 500 μg of (+)-disparlure in twine dispensers (Scentry Biologicals, Inc., Billings, MT, USA) and hung on tree trunks at a height of 1.5 m. Traps were checked at least 3 days after a release to ensure converged trap catches, meaning that the value did not increase with increased trapping time [33].

### 2.1. Experimental Design

Previous studies aimed at quantifying average proportion of target insects caught across the full sampling area of a trap have favored a single-trap, multiple-release design (Figure 1A). Here we used a single-release, multiple-trap design (Figure 1B). The rationale was that sample size for catch at each distance would be raised given that the number of insects available to us for release was limited. A requirement of this new approach was that traps must be spaced at a distance where they do not compete significantly. The minimal permissible distance for avoiding significant competition between milk carton traps of gypsy moth males was previously documented at about 40 m [34]. Both plot designs accommodated shifting wind directions.

Experiments were conducted in 2015, 2016, 2019, and 2020. In each year, we placed traps at various distances from the release point along the cardinal directions as shown in Figure 1B. Releases ranged from 50 to 500 males per release point at each time of release (Table 1). We used 3- to 7-day intervals between male moth releases to allow males adequate time to find traps [33].

In years 2015 and 2016, we established one plot and moved traps around the single release point to achieve specific distances; the number of releases at each distance ranged from 2 to 9 (Table 1). In 2019, we established three plots; in each plot, we moved traps to achieve specific distances and made three releases at each distance. The distance between plots was ≥2000 m to prevent interference. In 2019 and 2020, we deployed a total of six USDA milk carton pheromone-baited traps and released males next to a trap to estimate trap catch at 0 m. We made 20 releases for a total of 190 gypsy moth males; releases ranged from 3–20 males/release. Multiple plot locations allowed for averaging of local variations of the relevant conditions.

In 2019, we estimated average plume reach of a single USDA milk carton pheromone-baited trap using the indirect approach of Miller et al. [28]. We established three 100 × 100 m plots each separated by ≥2000 m to prevent interference. Fifty male moths at a time were released at 15, 30, 45, 60, and 75 m in each of the four cardinal directions (200 per distance) from a single trap (Figure 1A). Male moths released at each distance bore a unique fluorescent dye (DayGlo^®^, Cleveland, OH, USA) visible under black light [35]. Releases were made once or twice a week to minimize catch overlap; traps were checked and emptied at the time of release. A total of 9 releases/distance/plot occurred over 7 weeks and data collection ceased 3 days after the final release.

The fraction of males caught per trap collected from experimental design Figure 1B requires an additional conversion to be compared directly with the same fraction based on the trap set up shown in Figure 1A. Specifically, to interpret the results from Figure 1B, we assume that fractions of males caught by each of the four traps are uncorrelated, that is the fraction caught by each trap is independent of whether the three other traps are present. In this case, (fraction of males caught per trap) = ¼ × (total of males released)/(total of males caught). In practice, we assume no correlation between the traps if the distance between the traps is much larger than the plume reach D. For gypsy moth, this assumption is only approximate for the smallest release distances of r = 25 and 50 m, but for consistency, we nevertheless use (fraction of males caught per trap) = ¼ × (total of males released)/(total of males caught) for all of the data points resulting from Figure 1B set up.

### 2.2. Estimating Plume Reach and Catch Probability using Existing Methods

The existing approach of Miller et al. [28] assumes that: (1) insects displace by correlated random walks before they contact a pheromone plume, and as such they will quickly become randomly distributed even though their starting populations may have been clumped; (2) capture probability falls with distance by some starting probability of capture at the trap; (3) catch contribution from an annulus of area away from a trap is given by catch probability from that given distance multiplied by the number of insects inhabiting that annulus area; (4) overall catch probability (*T_fer_*) for a trap’s sampling area can be calculated from a catch probability profile across distance that is measured in release-capture experiments; (5) the maximum for trapping (sampling) radius is given by the distance at which no catch is recorded when a goodly number of insects have been released therefrom; and (6) an estimate of insects per trapping area is given by dividing catch number by overall catch probability so as to generate an estimate of absolute density. Average plume reach was estimated from the slope of a plot of distance of release vs. the inverse of average proportion caught per distance (MAG plot; Figure 2). Previous studies [28] found plume reach to be 20%–30% of trap diameter value (*L*), where:(1)$$L=\;\frac{2\pi }{MAG\;plot\;slope}$$

To estimate the sampling area of the trap, we used the maximum trapping radius, which is the farthest distance yielding a capture [28,30].

To estimate catch probability, we used the procedure developed by Miller et al. [28] and calculated annuli for each release distance, catch probability for each annulus and the product of annulus and a corresponding catch probability (Table 2). The overall catch probability across the sampling area (*T_fer_*) is calculated as mean of annulus area × catch probability per given annulus divided by mean annulus area, and can be used to estimate the absolute number of insects per trapping area as catch per one trap divided by the overall catch probability (*T_fer_*).

### 2.3. The Novel Model and Its Derivation

#### 2.3.1. Preliminaries and Definitions 

Our goal was to propose a simple and mechanistically sensible equation for the probability (*spT_fer_*(*r*)) with which a pheromone-baited trap captures a single male released distance *r* away from the trap. By *catch* we mean *converged catch*, which in the current type of experiment means we seek the $\lim_{t\to \infty }spT_{fer}\left(r\right)$, which we still call *spT_fer_*(*r*) for notational simplicity.

A key characteristic of the trap-male system is the so-called trap average plume reach, *D*, operationally defined as the maximum distance from the pheromone source (the trap) at which the male still shows some physiological response (the average plume reach is assumed isotropic, corresponding to the experimental design where it is determined from multiple uncorrelated measurements leading to any directionality averaging out). In reality, the response function is obviously not an infinitely sharp “yes” or “no” step function *p_0_ =*
*θ(r)*, but, rather, a smooth probability function p_0_ equal to 1 at *r = 0*, sharply decreasing at *r = D*, and zero at infinity (Figure 3). We assume that the insect release occurs at time *t = 0*, and the trap is left in place for a sufficiently long interval (48 h) for the male to be able to reach the trap—the converged catch—if released from any point up to *r = R_max_* away from the trap. Here *R_max_* = 1600 m is the longest “catch distance” experimentally recorded [20]; we assume that the probability for a male to cover a distance longer than *R_max_* over its lifespan is zero.

#### 2.3.2. Constructing the Model

Our overall approach is to explore limiting cases of small and large *r,* and then interpolate between them with a simple formula.
(1)In the limiting case of *r = 0*, the male is released right next to the trap. The insect is clearly well within the plume reach, and it becomes trapped with a constant probability *spT_fer0_*, which can be measured experimentally for the given trap type.(2)For small, but non-zero *r << D*, the plume reach is nearly as strong as at *r = 0*, and so the over-all catch probability should not be too much smaller than *spT_fer_(0)*. In this range, we expect the shape of *spT_fer_(r)* function to resemble *p_0_ (r)* of Figure 3. As the distance to the trap becomes comparable to the plume reach, *r ~ D*, *spT_fer_(r)* begins to decrease appreciably.(3)The most interesting and complex regime corresponds to large distances between the release point and the trap, *r >> D*, but still smaller than *R_max_*. We consider the male trajectory to be essentially 2-dimensional in this case, confined to a relatively narrow (compared to r) zone between the ground and the tree line. There are three distinct outcomes of a male trajectory at these distances (Figure 4). First, the male trajectory can enter the plume circle around the trap, and so the male becomes trapped with the probability *spT_fer_(0)*. Second, the male can travel outside of the circle of radius *r* around the release point; we assume that for large *r*, its likelihood of coming back and eventually getting trapped is negligible. If these were the only two possibilities, the proportion of males trapped would be ~($\frac{D}{r}$), which is the ratio of the total outward insect flux 2*πr* to the flux through the plume reach circle 2*πD*. Here, flux = (number of males) × (circle perimeter). However, at large values of *r*, that is, at large times elapsed from the release, the males begin to die or stop the search for various reasons, that is the insect flux through the “outer” circle *r* is not conserved, but instead decreases with time, and, hence with the distance from the release point. As the simplest approximation, we assume that male flux falls of as 1/*r*. The net result is that the proportion of males trapped decreases with distance as ~${\left(\frac{D}{r}\right)}^{2}$ for large *r* (the extra “*D*” makes sure that the expression is dimensionless). Thus, far away from the release point, the probability of the male trapped is $spT_{fer}\left(0\right){\left(\frac{D}{r}\right)}^{2}$.(4)Finally, for *r > R_max_*, *spT_fer_(r)* = 0, which means males simply do not travel that far.

Based on the above, we propose Equation (2) for *spT_fer_(r)*; one can verify that it satisfies all four of the limiting cases described above.
(2)$$spT_{fer}\left(r\right)=\left\{\begin{array}{r} \frac{spT_{fer}\left(0\right)}{1+{\left(\frac{r}{D}\right)}^{2}},\;r\leq R_{max}\\ 0,\;r>R_{max} \end{array}\right.$$

Note that when *r = D* exactly, that is the male is released right at the edge of the plume reach, *spT_fer_(D)* = $\frac{1}{2}$*spT_fer_*(0). This is because there is a 1/2 probability that, being right at the edge of the plume reach, the male flies in and, once inside the plume reach, gets trapped with the probability *spT_fer_*(0). The male can also fly in the opposite direction and escape the trap with probability 1/2. This behavior of *spT_fer_(r)* in Equation (2) is consistent with the meaning of plume reach (Figure 3).

Estimating the plume reach (*D*) from experimental data

Assuming ${\left(\frac{D}{r}\right)}^{2}$ >> 1, Equation (2) reduces to a pure power law decay:
$${\mathrm{spT}}_{\mathrm{fer}}\left(r\right)=\left\{\begin{array}{r} \frac{{\mathrm{spT}}_{\mathrm{fer}}\left(0\right)}{{\left(\frac{r}{D}\right)}^{2}},\;r\leq R_{\max}\\ 0,\;r>R_{\max} \end{array}\right.$$

Since Equation (2) is derived based on the long-range asymptotic considerations, a fit to the long-range part of the data is a better way of estimating *D* as compared to a fit to all of the data point (to be presented in Results for validation of the model across the entire range of distances).

From our estimate of *D* based on the MAG plot [28], the condition ${\left(\frac{D}{r}\right)}^{2}$ >> 1 should be satisfied for *r* ≥ 80 m.

We used a log-log plot (Figure 5) to find the value of plume reach *D*:(3)$$\ln(spT_{fer}\left(r\right))=\;-2\ln\left(r\right)+\ln(D^{2}\times spT_{fer}\left(0\right))$$

We then used JMP 11 Pro [36] to fit Equation (3) to the experimental data points ≥80 m (Figure 5), which yielded D = 25.6 ± 3 m, where the error margin is estimated from the uncertainty of the fit line.

#### 2.3.3. Estimating the Absolute Insect Density from Individual Trap Catch Data 

Here, we address the following question: given a specific number of males M caught by the pheromone trap over the typical observation time, what is the most probable male population density $\bar{\rho_{mp}}$ in the surrounding area? Further, what are the lower and upper bounds for this value, within specified confidence interval *p*? A single male distance r away from the trap contributes *spT_fer_(r)* to the total number of males caught by the trap. Thus, assuming that a 2D male population density *ρ*(*x*,*y*), the total number of males caught is given by the integral over the area of interest [28]:(4)$$\bar{M}=\iint spT_{fer}\left(r\right)\rho \left(x,y\right)dxdy=2\pi \int_{0}^{R_{max}}spT_{fer}\left(r\right)\rho \left(r\right)rdr$$
where we have taken the area to be bounded by the maximum possible male flight distance *R_max_*—it is natural to assume that no male can reach the trap from outside of the circle of radius *R_max_* over the trapping period. Note that the case of a single male at position *r*_0_ can formally be represented by the delta function density *ρ*(*x*,*y*) = δ(*r* − *r*_0_) substituting it into Equation (4) yields $\bar{M}$ = *spT_fer_*(*r*_0_), as it should. As a side note, Equation (4) implies a certain limitation on the asymptotic behavior of *spT_fer_(r)* for $\frac{D}{r}$ << 1, as the integral has to converge at the upper limit, which in turn implies that if *spT_fer_(r)*~${\left(\frac{D}{r}\right)}^{b}$ for large *r*, then *b* cannot be less than 2.

To proceed with the derivation of bounds on the male density, assume that *ρ*(*r*) does not have any systematic variation over the collection area, in which case we can replace *ρ*(*r*) with its average:(5)$$\rho \left(r\right)=const=\bar{\rho }$$

With this, we apply Equation (2) to obtain:(6)$$\bar{M}=2\pi \bar{\rho }spT_{fer}\left(0\right)\int_{0}^{R_{max}}\frac{rdr}{1+{\left(\frac{r}{D}\right)}^{2}}=\pi \bar{\rho }spT_{fer}\left(0\right)D^{2}ln\left(1+{\left(\frac{R_{max}}{D}\right)}^{2}\right)$$
and
(7)$$\bar{\rho }=\left(\frac{\bar{M}}{spT_{fer}\left(0\right)}\right)\times \frac{1}{\pi D^{2}ln\left(1+{\left(\frac{R_{max}}{D}\right)}^{2}\right)}$$

To relate the average $\bar{\rho }$ in the above equation to actual integer number of males caught *M*, we need to make an assumption about the kind of statistical distribution that is appropriate for *M*. We argue that the Poisson distribution (with the expected value $\bar{M}$) is most appropriate: it implies that the density of males does not change significantly over the trapping period due to the action of the trap, which is a reasonable assumption for a single trap in a large open area. Using the desired confidence interval *p*, the lower (min) and upper (max) bounds on the average male catch $\bar{M}$ [37]:(8)$$\frac{1}{2}\;\chi^{2}\;\left(\frac{\alpha }{2};2M\right)\leq \;\bar{M}\;\leq \frac{1}{2}\;\chi^{2}\;\left(1-\frac{\alpha }{2};2M+2\right)$$
where *α* = 1 − *p*, and *χ*^2^(*q*,*n*) is the quantile function (corresponding to a lower tail area *q*) of the chi-squared distribution with *n* degrees of freedom. Introducing, for notational simplicity:$$\mu =\left(\frac{1}{spT_{fer}\left(0\right)}\right)\times \frac{1}{\pi D^{2}ln\left(1+{\left(\frac{R_{max}}{D}\right)}^{2}\right)}$$

We arrive at the desired lower and upper bounds on the average density $\bar{\rho }$
(9)$$\frac{\mu }{2}\;\chi^{2}\;\left(\frac{\alpha }{2};2M\right)\leq \;\bar{\rho }\;\leq \frac{\mu }{2}\;\chi^{2}\;\left(1-\frac{\alpha }{2};2M+2\right)$$

For *M* males caught, the expected value (average) of the Poisson distribution that maximizes the probability is *M* itself; therefore, with *M* males caught, the most probable (average) male density in the collection area is:(10)$$\bar{\rho_{mp}}=\left(\frac{M}{spT_{fer}\left(0\right)}\right)\times \frac{1}{\pi D^{2}ln\left(1+{\left(\frac{R_{max}}{D}\right)}^{2}\right)}=µM$$

To convert the male density to number of males per ha, and assuming *D* and *R_max_* are given in meters, *μ* in Equations (9) and (10) needs to be multiplied by 10,000.

## 3. Results

### 3.1. Estimating Plume Reach and Catch Probability Using Existing Methods

To estimate plume reach of a USDA milk carton pheromone-baited trap, we released a total of 8600 adult males and captured 1289 (15%). Capture rates ranged from 2% to 45%. We estimated plume reach to be in the range of 19–28 m.

Since, in previous studies, 1 male was caught 1600 m away from the release point [20], we used the maximum trapping radius of 1600 m (Figure 6), which yields a trapping area of ~804 ha. The overall catch probability (*T_fer_*) as calculated by the methods of Miller et al. [28] was 0.0025.

### 3.2. Validation of the New Model

Our main result is Equation (2), see Section 2.3, which relates proportion of males caught in a pheromone-baited trap (*spT_fer_*) to distance (*r*) from the trap and plume reach (*D*). The equation predicts steep, power law-like decline of trap catches with increasing distance from the trap (Figure 7). The model gives the probability (*spT_fer_* (r)) of catching a single male located at a distance *r* from the trap with a plume reach (*D*). Here, *spT_fer_(*0*)* is the probability of catching a male located in the immediate vicinity of the trap (*r* = 0). This value is determined experimentally as described in Section 2.1 and is equal 0.37. *R_max_* is the distance from the trap, beyond which trap catch is always 0. In the case of gypsy moth, *R_max_* = 1600 m [20]. The only fitted parameter of Equation (2) is plume reach *D* which was obtained as described in the Methods. The fitting was performed for large values of *r*, yielding the plume reach *D = 25.6 ± 3 m*, therefore, Equation (2) for gypsy moth is:
(11)$$spT_{fer}\left(r\right)=\left\{\begin{array}{r} \frac{0.37}{1+{\left(\frac{r}{25.6}\right)}^{2}},\;r\leq 1600\;meters\\ 0,\;r>1600\;meters \end{array}\right.$$

Within the model, the overall catch probability across the sampling area (*T_fer_*) can be calculated as $\bar{M}$ using Equation (6), where one assumes $\bar{\rho }=\frac{1}{{\pi R_{max}}^{2}}$, which is the average density of 1 male over the entire trapping area:(12)$$\bar{M}=spT_{fer}\left(0\right){\left(\frac{D}{R_{max}}\right)}^{2}ln\left(1+{\left(\frac{R_{max}}{D}\right)}^{2}\right)$$

The corresponding $\bar{M}=\;T_{fer}=0.0008$.

The model is verified on all of the data points available to us (Figure 7), including small values of r < 80 m, not used to obtain the plume reach value. The correlation is R = 0.9 over the entire data set. It is noteworthy that even though the model was derived using asymptotic considerations for r >> D, it still agrees well with experimental values when *r* is small.

### 3.3. Model Application: Estimate of the Average Gypsy Moth Density from the Catch Data

We have applied the gypsy moth-specific model (Equation (11)) to estimate lower and upper bounds of moth density based on moth counts in USDA milk carton pheromone-baited traps (Figure 8, Table 3). We stress that these counts correspond to converged catches (≥3 days). The details of the experimental design are presented in Section 2.1.

## 4. Discussion

The new predictive relationship established between proportion of males caught in a pheromone-baited trap (*spT_fer_*), distance (*r*) from the trap, and plume reach (*D*) yields a plume reach of 25.6 ± 3 m; the proposed equation provides a straightforward way to estimate absolute density from the trap catch. The corresponding estimate of plume reach is remarkably close to the estimate obtained using the methodology developed by Miller et al. [28]; both agree with observations made by other researchers. Elkinton and Cardé [34] observed interactions between traps spaced 40 m apart, indicating that plume reach of a milk carton pheromone-baited trap is ≥20 m. Another study reported wing fanning of a much smaller proportion of gypsy moth males at 40 m from the pheromone source [38].

The estimates of the overall catch probability (*T_fer_*) obtained using the existing method [28] and the proposed model are 0.0025 and 0.0008, respectively. We explain the 3-fold difference as follows. First, the existing method converts the integral of Equation (4) into a partial Reimann sum, coarse-grained by the use of discrete annuli (see Methods), while the new method directly integrates over the smooth curve, Equation (6), representing *T_fer_(r)* as a function of *r*. Within the existing method based on discrete equidistant annuli (in our case d = 50 m apart), the annuli that do not contain experimental data points are skipped in the calculation. In the case of our gypsy moth data, the skipped annuli are mostly the annuli corresponding to large distances from the trap, where *spT_fer_(r) × (annulus area) ~ 2Pi (D/r)^2^* × *r* × *d* is low, and further decreases with *r*. Thus, the contributions to *T_fer_* from low *spT_fer_(r) × (annulus area)* values are underrepresented in the final tally, leading to the *T_fer_* being overestimated compared to the integral over the entire region from *r = 0* to *r = R_max_*, Equation (6). By skipping a number of annuli corresponding to large *r* values, the existing method effectively uses lower *R_max_*: clearly, the probability to catch one male from a smaller area is larger (see also Equation (12)). The above logic explains the overestimation of *T_fer_* compared to the value obtained by directly integrating the model in Equation (6). Note that for an insect, for which fine-grained data points for *spT_fer_(r)* are available [32] for the entire span of *r* values from near 0 to *R_max_*, no consistent overestimation of *T_fer_* using the existing approach [28] is expected.

To assess possible influence of the discretization interval (d = 50 m used here) on the estimate of *T_fer_*, we calculated *T_fer_* using the partial Reimann sum approximating Equation (6) in two ways: first, taking *T_fer_(r)* value (Equation (2)) at the left side of each discretization interval (annulus), then taking it to be on the right side of it. We used the same set of annuli as previously. The resulting difference in the estimated *T_fer_* is 3-fold, from 0.00373 to 0.00147, indicating that the discretization into 50-m annuli—limited in practice by the extremely laborious process of obtaining experimental data points at each *r*—may still be too coarse for a converged estimate of *T_fer_* for gypsy moth using the existing method based on coarse-graining the integral. Since the characteristic decay length of *spT_fer_(r)* is the plume reach *D* (Equation (2)), we suggest that a highly converged estimate relying on a discrete sum to approximate the true integral should use a discretization interval of much less than *D*, e.g., *d* = 10 m for gypsy moth. However, obtaining high quality relevant experimental data for gypsy moth, for a set of, say, 160 distance points, *d* = 10 m apart from 0 to *R_max_* = 1600 m would be extremely difficult in practice: the 18 existing data points reported here took the team four seasons to collect. This is yet another strong motivation for the need for a single equation for *spT_fer_(r)* valid for all values of *r*, such as Equation (6). Note that the value of *T_fer_* obtained within the proposed model represents the best fit over the entire set of experimental points, thus the outliers, especially at low distances (*r*) from the trap (Table 2), have much less influence on the final value of *T_fer_*.

However, despite being different, both values of *T_fer_*—the one obtained with the discrete annuli summation and the one based on Equation (12)—agree with previous findings in that in most insects the overall catch probability is very low, <0.02, which can hinder detection of low-density populations as well as lead to underestimation of the detected population’s density [29]. We emphasize that a very low value for *T_fer_* most likely results from a highly vagile insect target rather than a poorly performing trapping system [29].

Bau and Cardé [26] used simulation models of odor dispersal and plume acquisition to predict the probability of detection of low-density gypsy moth populations. Their results indicated that at 30 males/km^2^ (=0.3/ha), a probability of false negative was high and that trap efficiency had a profound effect on detectability. Our results agree with this conclusion. We found that a catch of 0 can result from non-zero density of up to six gypsy moth males/ha (Table 3). Nevertheless, the USDA milk carton pheromone-baited trap appears to be sensitive enough to provide meaningful lower and upper bounds on absolute population density from a catch of 1 male/trap. These bounds can be used in management programs. Note that the model gives no prediction of the gypsy moth population location, only the population density in the given area; therefore, further delimitations are needed to determine exact location and extent of the population [39,40]. This is especially important in the uninfested areas, where gypsy moth populations are sparse [33]. Currently, the National Gypsy Moth Slow the Spread Program utilizes previously optimized 2-km and 3-km grids of USDA milk carton pheromone-baited traps for detecting isolated colonies [39,41,42] and 1000-m to 500-m grid for delimiting populations for treatment planning or evaluation of a previously applied treatment [6,39,43,44]. Our results indicate that these fairly coarse grids are sufficient for detection and delimitation due to plume reach and sensitivity of a milk carton trap.

Understanding the relationship between trap catches and absolute density is especially important at very low catches. Management programs rely on pheromone-baited traps to make decisions on the necessity and type of treatment application. Depending on the goal and available resources, ability to calculate the absolute density would allow for adjusting the response to be more or less conservative. However, estimated bounds cannot be directly applied to season-long trap catches because the population density changes in time according to Gaussian distribution [45]. Instead, season-long population density can be used to first estimate abundance during peak flight [45]. Then, the absolute population density during peak flight can be estimated using Equation (11). This would enable a determination of the most appropriate method of control based on the location of the population and the goal of the control program. For example, if the goal is eradication, absolute maximum population density can be used to assign a more aggressive treatment option.

## 5. Conclusions

One of the key outcomes of this work is a simple mathematical relationship between the probability of trap catch and the distance between an insect and a trap. The relationship derived applies to converged catch, which means that the trapping time is long enough that longer exposure does not lead to greater catch. This relationship can be used to conduct further analyses to improve and optimize monitoring in management programs; for example, one can estimate a probability of a localized infestation for the given trap catch data obtained using regular monitoring grid. Such analysis might lead to a more economical grid size than currently used. Moreover, the simplicity of the obtained relationship hints at the possibility that it may be general and applicable to other insect species. We plan to test this conjecture in a future study.

## Figures and Tables

**Figure 1 insects-11-00673-f001:**
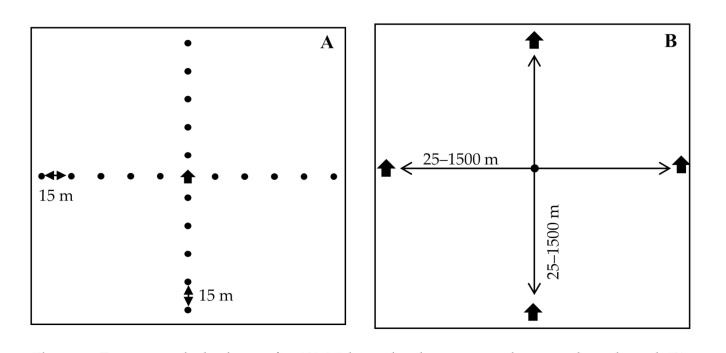
Experimental plot layout for (**A**) Male moth release point, plume reach study and (**B**) Pheromone-baited trap, absolute density study.

**Figure 2 insects-11-00673-f002:**
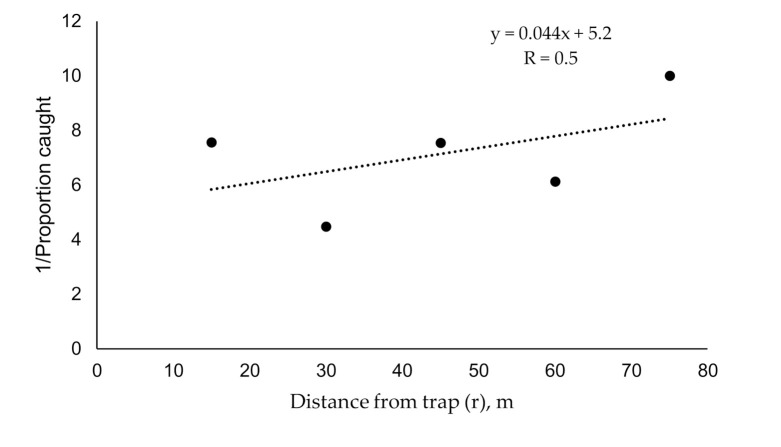
Inverse of proportion of gypsy moth males caught in pheromone-baited traps released at various distances from traps (r) (MAG plot transformation [28]). The proportion was calculated as an average over nine catches for each distance.

**Figure 3 insects-11-00673-f003:**
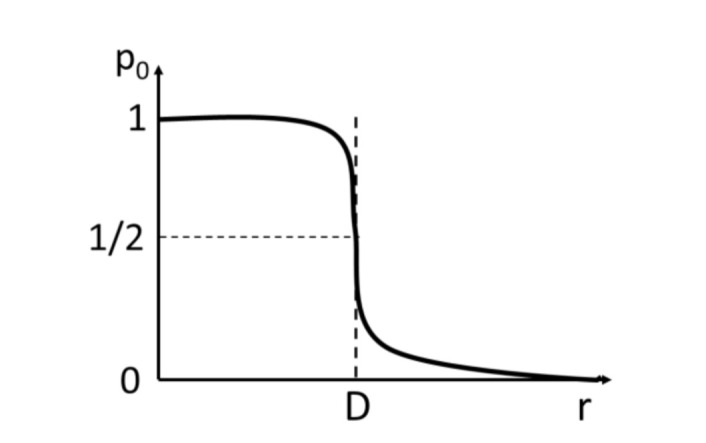
Probability p_0_ of the male response to a pheromone trap distance r away. D is the plume reach.

**Figure 4 insects-11-00673-f004:**
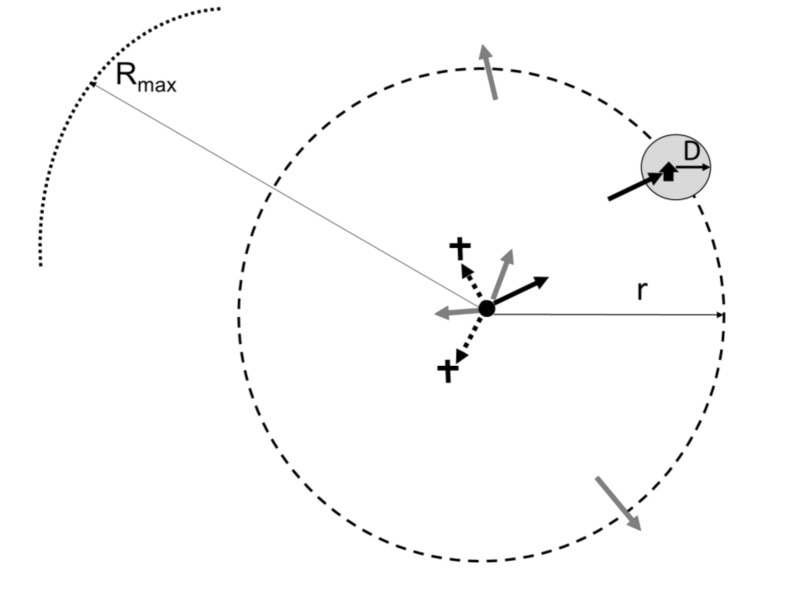
Outflux of insects through an imaginary circular boundary of radius r >> D surrounding the release point (black circle in the middle). Not every released male (indicated by five arrows at the center) reaches the boundary (solid black arrow)—some trajectories terminate (dotted arrows and crosses) inside the circle, while some trajectories continue outside the circle (grey arrows). Of those that do reach the boundary at r (solid black arrow), the fraction caught by the trap is the fraction of the total insect flux through the circle of radius D (plume reach).

**Figure 5 insects-11-00673-f005:**
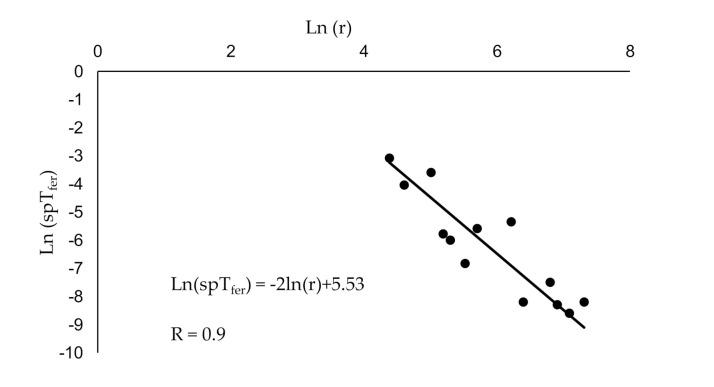
Log-log plot of proportion males caught in pheromone-baited traps located at various distances (r) from male moth release points. Black line is the least square fit to the experimental data points.

**Figure 6 insects-11-00673-f006:**
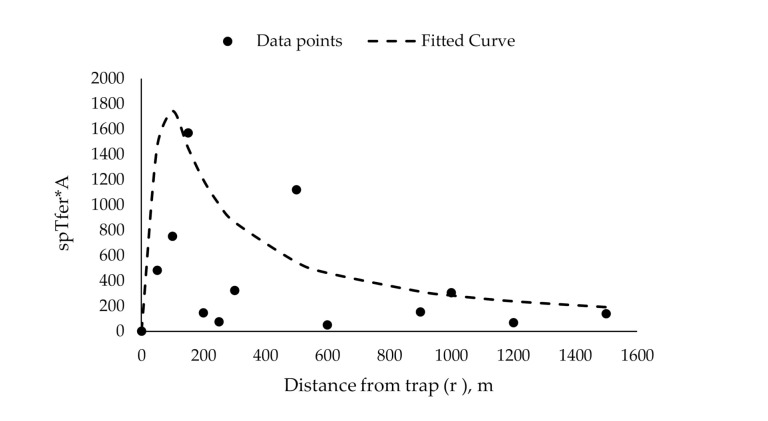
Proportion of released gypsy moth caught times annulus (*spT_fer_* × A) at various distances from traps (r) (Miller plot transformation, [28]).

**Figure 7 insects-11-00673-f007:**
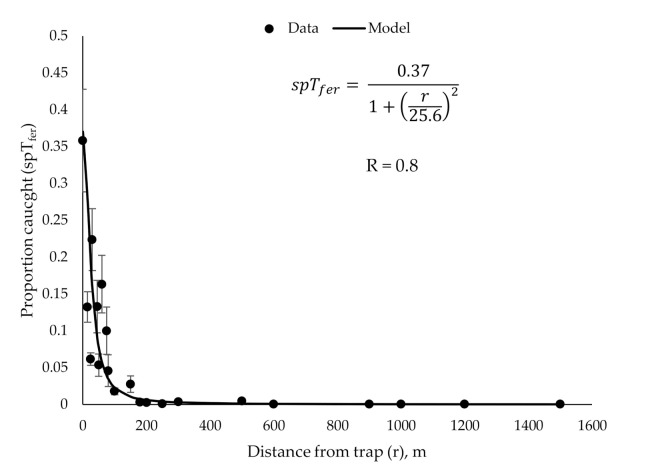
Experimental validation of the new model. Dots: experimental proportion of males caught in pheromone-baited traps placed at various distances from the release point (±SEM). Error bar is not shown when smaller than the symbol size. Solid line: the model described by Equation (2).

**Figure 8 insects-11-00673-f008:**
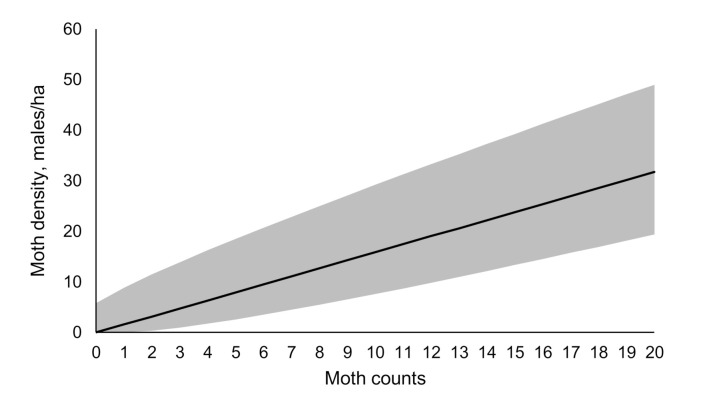
Estimate, based on the new model, of absolute gypsy moth male density from catch in United States Department of Agriculture milk carton pheromone-baited traps. Grey area indicates the range between lower and upper bounds with 95% probability, black line in the middle indicates the most probable density $\bar{\rho_{mp}}$ for each specific trap catch.

**Table 1 insects-11-00673-t001:** Releases of male gypsy moths at various distances from pheromone-baited traps in plots with single-release, multiple-trap design (Figure 1B).

Year	Distance (m)	Number of Males Released	Number of Releases
2015	25	200	9
	50	200	3
	80	200	3
	100	200	3
	250	200	5
	500	200	5
	1000	200	5
2016	25	200	8
	100	200	6
	150	200	2
	180	200	2
	200	200	8
	250	200	3
	300	200	3
2019	300	50	9
	600	100	9
	900	200	9
	1200	300	9
	1500	500	9

**Table 2 insects-11-00673-t002:** Catch probability at various distances from male gypsy moth release point. Catch probability is calculated based on the data reported here using existing methods [28].

Release Distance [r] (m)	Annulus Area [A] (m^2^)	Catch Probability for Each Annulus [spT_fer_(A)]	Annulus Area × Catch Probability [spT_fer_(A) × A]
0	0	0.37	0
50	7854	0.24	1858
100	23,562	0.16	3855
150	39,270	0.16	6283
200	54,978	0.01	676
250	70,686	0.004	295
300	86,394	0.0008	72
500	149,226	0.03	4477
600	180,642	0.003	582
900	274,889	0.004	1222
1000	306,305	0.002	511
1200	369,137	0.007	2461
1500	463,385	0.01	4548

**Table 3 insects-11-00673-t003:** Estimates of lower and upper bounds, and the most probable absolute gypsy moth male density $\bar{\rho_{mp}}$ (males/ha) corresponding to catches in United States Department of Agriculture milk carton pheromone-baited traps.

Catch	Lower Bound	Upper Bound	Most Probable Density
0	0	5.8	0
1	0.04	8.8	1.6
2	0.38	11.5	3.2
3	0.98	13.9	4.8
4	1.7	16.3	6.4
5	2.6	18.5	7.9
6	3.5	20.7	9.5
7	4.5	22.9	11.1
8	5.5	25	12.7
9	6.5	27.1	14.3
10	7.6	29.2	15.9
11	8.7	31.2	17.5
12	9.8	33.3	19
13	11	35.3	20.6
14	12.2	37.3	22.2
15	13.3	39.3	23.8
16	14.5	41.2	25.4
17	15.7	43.2	27
18	16.9	45.2	28.6
19	18.2	47.1	30.2
20	19.4	49	31.7
21	20.6	51	33.3
22	21.9	52.9	35
23	23.1	54.8	36.5
24	24.4	56.7	38.1
25	25.7	58.8	39.7
26	27	60.4	41.3
27	28.2	62.4	42.9
28	29.5	64.2	44.4
29	30.8	66.1	46
30	32.1	68	47.6
31	33.4	69.8	49.2
32	34.7	71.7	50.8
33	36.1	73.6	52.4
34	37.4	75.4	54
35	38.7	77.3	55.6
36	40	79.1	57.1
37	41.4	81	58.7
38	42.7	82.8	60.3
39	44	84.6	61.9
40	45.4	86.5	63.5
41	46.7	88.3	65.1
42	48	90.1	66.7
43	49.4	91.9	68.3
44	50.7	93.8	69.8
45	52.1	95.6	71.4
46	53.5	97.4	73
47	54.8	99.2	74.6
48	56.2	101	76.2
49	57.5	102.8	77.8
50	58.9	104.6	79.4

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
