# Peer review of "Bounds on Absolute Gypsy Moth (Lymantria dispar dispar) (Lepidoptera: Erebidae) Population Density as Derived from Counts in Single Milk Carton Traps"

_insects, 2020, doi:10.3390/insects11100673_

Round 1

Reviewer 1 Report

It was a pleasure to read the article titled "Bounds on absolute gypsy moth (Lymantria dispar dispar) (Lepidoptera: Erebidae) population density as derived from counts in single milk carton traps" submitted by Onufrieva et al. as the present a well written manuscript that will have great impact on the study of gypsy moth dynamics. I have only minor suggestions, though there does seem to be a problem with the citation management software the authors used (e.g., line 112), and the citations will need to be updated prior to publication.

Minor Comments:

In Table 2, why does spTfer(A) occassionally increase with increasing distances?

Lines 290-293. Please explain how these numbers can be so drastically differeny (plume length vs. detection distance)

Figure 6. Two datapoints (r ~180m and r ~ 550m) appear to be extreme outliers. Do your results change with their exclusion?

Reference 22. Change to sentence caps.

Author Response

Thank you very much for taking the time to read the manuscript and for your valuable comments. Below are responses to specific comments and concerns:

  1. In Table 2, why does spTfer(A) occasionally increase with increasing distances?

This reflects variability between trap catches, in Figure 7 we show standard errors associated with trap catch at each distance.

  1. Lines 290-293. Please explain how these numbers can be so drastically different (plume length vs. detection distance)

Detection distance depends on several factors, including plume reach and the maximum distance an insect is able to travel (Rmax) and the latter can be many times larger than the plume reach.

  1. Figure 6. Two data points (r ~180m and r ~ 550m) appear to be extreme outliers. Do your results change with their exclusion?

We tried excluding both distances, but the results didn’t change significantly. Since this data were obtained legitimately and we have no valid reason to exclude them, we decided to keep both data points.

  1. Reference 22. Change to sentence caps.

Reference changed, thank you for catching this.

Reviewer 2 Report

Comments to the manuscript ID: insects-949702.

Title: Bounds on absolute gypsy moth (Lymantria dispar dispar) (Lepidoptera:

Erebidae) population density as derived from counts in single milk carton traps.

Authors: Ksenia Onufrieva *, Alexey V. Onufriev, Andrea Hickman, James Miller.

General comments:

Please, fix citations at number of places, for example Lines 112-113, 140-141.

Figures 2-8 are not referred in the text.

Please be consistent using the figures axes title style. For example Fig. 6 title starts with regular character while in Figs. 7 and 8 capital characters are used.

Line 36. Author name who describe the species is missing.

Line 45. Change to (+)-disparlure

Line 84. “and hung at 1.5 m.”
Please indicate that the measurement is a height. Were traps hung on host-plants?

Line 114. “Male moths released at each distance bore a unique fluorescent dye (DayGlo®, Cleveland, OH) visible under black light [35]. Actually in the publication by Tcheslavskaia, K.S., et al 2005, there is no “black light” method mentioned. Authors used microscope with UV light to detect the presence of fluorescent powder on wings, antennae or body.

Figure 2. “…proportion caught in pheromone-baited traps…” is the value of proportion an average? Legend of the figure has to be self-explanatory, hence, please indicate the number of replicates used.

Figure 4. What is the meaning of different arrow style? Please provide colour image or remove the words “(red star)” from the legend. Actually, there is no star in the figure. What do crosses indicate?

Figure 6. X axis missing distance units. Good example Fig. 7.

Author Response

Thank you very much for taking the time to read the manuscript and for your valuable comments. Below are responses to specific comments and concerns:

  1. Please, fix citations at number of places, for example Lines 112-113, 140-141.

We fixed citations, thank you for catching.

  1. Figures 2-8 are not referred in the text.

We double-checked and the figures are referenced in the text – please, see lines 141 (Fig. 2), 169 (Fig. 3), 190 (Fig. 4), 234 (Fig. 5), 292 (Fig. 6), 300 (Fig. 7), 325 (Fig. 8).

  1. Please be consistent using the figures axes title style. For example, Fig. 6 title starts with regular character while in Figs. 7 and 8 capital characters are used.

The axis title in Fig. 6 is fixed to match other figures’ styles.

  1. Line 36. Author name who describe the species is missing.

Author name included

  1. Line 45. Change to (+)-disparlure

Corrected per reviewer’s request

  1. Line 84. “and hung at 1.5 m.” Please indicate that the measurement is a height. Were traps hung on host-plants?

We changed the sentence to read “and hung on tree trunks at a height of 1.5 m”

  1. Line 114. “Male moths released at each distance bore a unique fluorescent dye (DayGlo®, Cleveland, OH) visible under black light [35]. Actually, in the publication by Tcheslavskaia, K.S., et al 2005, there is no “black light” method mentioned. Authors used microscope with UV light to detect the presence of fluorescent powder on wings, antennae or body.

Black light is a kind of UV light so there was no contradiction. We changed “black light” to “UV light” for consistency, and we confirm that the same method was used in both papers.

  1. Figure 2. “…proportion caught in pheromone-baited traps…” is the value of proportion an average? Legend of the figure has to be self-explanatory, hence, please indicate the number of replicates used.

We added “The proportion was calculated as an average over 9 catches for each distance.” to the caption.

  1. Figure 4. What is the meaning of different arrow style? Please provide colour image or remove the words “(red star)” from the legend. Actually, there is no star in the figure. What do crosses indicate?

The legend is corrected to include descriptions of arrows and to replace the red star with the black circle (actually in the picture). The crosses indicate the trajectories of gypsy moth males terminated inside the circle, it is described in the legend.

  1. Figure 6. X axis missing distance units. Good example Fig. 7.

The X axis title is fixed to match figure 6, distance units added.